# Congenital Syphilis: A U.S. Perspective

**DOI:** 10.3390/children7110203

**Published:** 2020-10-29

**Authors:** Alvaro E. Galvis, Antonio Arrieta

**Affiliations:** 1Department of Infectious Diseases, Children’s Hospital of Orange County, Orange, CA 92868, USA; agalvis@choc.org; 2School of Medicine, University of California, Irvine, CA 92697, USA

**Keywords:** syphilis, *Treponema pallidum*, congenital infection

## Abstract

Congenital syphilis still represents a worldwide public health problem. If left untreated, it can lead to fetal demise and high neonatal morbidity and mortality. Unfortunately, in the last decade, there has been a resurgence of cases in the U.S. This review discusses the ongoing problem of this preventable congenital infection, vertical transmission and clinical manifestations while providing a guidance for the evaluation and management of infants born to mothers with reactive serologic tests for syphilis.

## 1. Introduction

Congenital syphilis (CS) occurs when the *Treponema pallidum* subspecies, *pallidum*, infects the fetus of a woman typically affected by primary or secondary syphilis. Despite wide understanding of the disease, ability to treat and optimal preventive strategies, CS remains a major cause of fetal and neonatal mortality globally. The global burden of CS is exacerbated by the high prevalence of co-infection with human immunodeficiency virus in adults.

Mother to child transmission (MTCT) can occur at any time during gestation; risk of transmission in relation to the maternal stage of infection is highest during secondary syphilis while fetal infection during late latent infection is uncommon. In 1988, the surveillance case definition for national reporting of CS was broadened by the Centers for Disease Control and Prevention (CDC) to include: (1) a condition affecting stillbirths and infants born to mothers with untreated or inadequately treated syphilis regardless of signs in the infant or (2) a condition affecting an infant with clinical evidence of congenital syphilis including direct detection of *Treponema pallidum* or a reactive nontreponemal syphilis test with signs on physical examination, radiographs, or cerebrospinal fluid analysis [1].

CS is the second most common cause worldwide of preventable stillbirth [2]. It could also result in prematurity and low birth weight. It may be clinically apparent immediately after birth or it can remain asymptomatic for months or years. Newborns can be successfully treated with appropriate antibiotics started early after delivery, but this requires a high index of suspicion and appropriate diagnosis which may at times be difficult [3].

Testing during prenatal care (PNC) visits and timely treatment with appropriate antibiotics for maternal stage of syphilis will prevent most cases of CS. Still, close to a third of CS cases were diagnosed in newborns who’s mothers were tested during pregnancy and may have acquired infection after initial testing, emphasizing the need for multiple testing of pregnant women in areas of high prevalence of primary and secondary syphilis or in women at high risk for acquiring infection [4].

Focus on prevention of CS is important but it must start with understanding of the socio-economic circumstances that place young women at risk for acquiring syphilis. Poverty is the common denominator for homelessness, drug use, exchange of sex for drugs or money, incarceration and low education level; all features for increased risk for syphilis. Efforts to overcome these inequities are under way, meanwhile, focus on expanding care for pregnant women and understanding the missed opportunities for prevention and implementation of interventions tailored to local experience will help halt the continued increase in CS.

## 2. Epidemiology

Worldwide, more than half a million cases of CS were diagnosed in 2016 (rate 472 cases per 100,000 live births) resulting in over 200,000 stillbirths and early neonatal deaths [5]. Estimated adverse birth outcomes decreased slightly but not statistically significantly from 397,000 cases in 2012 to 355,000 (which excluded 306,000 asymptomatic cases) in 2016 [5]. In 2007, the World Health Organization (WHO) launched a global initiative for the elimination of MTCT of syphilis and human immune deficiency virus (HIV) with three simple interventions: (a) reach at least 95% of at least 1 PNC visit, (b) 95% of women receiving PNC tested for syphilis and (c) at least 95% of women treated adequately and timely for syphilis.

Recently, the CDC reported that CS cases had increased each year in the United States (U.S.) since 2012 which, as expected, coincided with the increased rate of primary and secondary syphilis in women of childbearing age [6] (see Figure 1). In 2017, there were 918 babies born with CS in the U.S. and the national CS rate was 23.3 cases per 100,000 live births, the highest rate reported since 1997, an increase of 153.3% relative to 2013 and to 1306 in 2018 (increase of 261%) [4]. CS rates were 6.4 times and 3.3 times higher among infants born to black and Hispanic mothers (86.6 and 44.7 cases per 100,000 live births), respectively, compared to white mothers (13.5 cases per 100,000 live births). Significant ethnic, socioeconomic and geographic differences affect the risk for primary or secondary syphilis in women of childbearing age [4].

## 3. Biology of Treponema Pallidum

Syphilis is a chronic infection caused by *Treponema pallidum* subspecies *pallidum* (hence referred to as *T. pallidum*) which belongs to the family of spiral-shaped bacteria, the *Spirochaetaceace*, commonly referred to as the spirochetes. Other members of this order include the genera *Borrelia* and *Leptosira. T. pallidum* is related to other pathogenic treponemes which are all morphologically identical to each other and cause nonvenereal diseases, including *T. pallidum endemicum* (bejel), *T pallidum pertenue* (yaws), and *T carateum* (pinta), which can be differentiated from one another by their clinical manifestations and genetic differences [7,8].

The *T. pallidum* cellular structure is composed of a fragile outer membrane that lacks liposaccharides (LPS), an inner cytoplastic membrane and a thin layer of peptidoglycan between the membranes that provides structural stability. The outer membrane has also been implicated as a major virulent factor through a variety of outer membrane proteins that can serve from inhibiting optimization to facilitate adhesion with cell host surface proteins and basement membrane [9]. An excellent review about the outer membrane by Radolf and Kumar is available [9]. The endoflagella that allow for the characteristic corkscrew mobility are located in the periplasmic space [10].

The genome of *T pallidum* is 1.14 Mb, which is strikingly smaller than most conventional gram-negative and gram positive bacteria, such as *Escherichia coli*, which is 4.6 Mb and *Staphylococcus aureus*, which is about 2.8 Mb [11]. The lack of genetic sequences, therefore, leads *T. pallidum* to have a limited metabolic capacity and hence requires macromolecules from the host environment for most of its nutritional requirements. Analysis carried out by the Genome Sequencing Project demonstrated that *T. pallidum* is capable of carrying out glycolysis and carries the ability to interconvert amino acids and fatty acids. However, the organism lacks the ability for using alternative carbon sources for energy and de novo synthesis of enzyme cofactors or nucleotides. Instead, the genome encodes for the various transporters needed to obtain the vast majority of essential macromolecules [8,10].

## 4. Clinical Manifestations

As discussed above, CS occurs when syphilis during pregnancy goes untreated, is treated late (<4 weeks before delivery) or inadequately (incomplete regimen for stage of disease, treatment with an agent other than penicillin); it may result in death (6.5% of cases) of which more than 80% are stillbirths. Prematurity and low birth weight were strongly associated with death among neonates born with CS [12,13].

Most newborns born to mothers with untreated syphilis appear normal and may have no clinical or laboratory manifestations of CS [14]. Diagnosis and treatment decisions may be difficult and not free of controversy particularly with interpretation of recent establishment of reverse sequence algorithm (see diagnosis section) [15,16].

### 4.1. Early Congenital Syphilis

Early CS is diagnosed when children develop manifestations of disease within the first 2 years of life. These are typically systemic, reflecting the inflammatory process taking place at one or many of the organs that became affected during transplacental infection [3,17,18] (see Table 1).

Mucocutaneous lesions are common, affecting 40–60% of infants with CS. Characteristic rash is typically maculo-papular, prominently distributed on the palms and soles and similar to that of secondary syphilis, it evolves from erythematous to copper in color; *pemphigus syphyliticus* is unique to newborns and consists of vesicobulous lesions which may peel and eventually crust and heal with noticeable skin wrinkling; occasionally mucous lesions, which are elevated and flat, may develop in perioral and perianal regions and go unrecognized [19] (see Figure 2). Some infants may develop rhinitis, characterized by a copious nasal discharge, which may become purulent if untreated. Condylomatous and bullous lesions, as well as nasal discharge, contain large amounts of *T pallidum* and are highly contagious.

Liver and spleen are frequently involved with hepatosplenomegaly, a common feature on physical exams, which may take months to resolve after treatment. Liver enzymes, direct bilirubin and alkaline phosphatase are typically elevated. Hematological abnormalities are common; anemia is noticed in 75% of patients and 50% will have an elevated white blood cell count with monocytosis. Thrombocytopenia and petechia occur frequently and may be the only manifestation of CS. Skeletal manifestations are present radiographically in >95% of symptomatic and 25% of asymptomatic CS infants. Lesions are typically symmetric and involve primarily long bones.

Destruction of tribal tubercle (Wimberger sign) is highly suggestive of CS [20] (see Figure 3). Periostitis of the metaphysis of the long bones results in pseudoparalysis (pseudoparalysis of Parrot) and distinctive radiographic features (see Figure 3).

CNS involvement is common; cerebrospinal fluid (CSF) abnormalities consistent with aseptic meningitis are present in >50% of symptomatic and 10% of asymptomatic infants. It is often the only finding elicited during the work up for sepsis in young febrile infants [14]. Non treponemal tests should be considered in all febrile infants with CSF pleocytosis and no microbiological diagnosis.

Other manifestations of CS include pulmonary involvement (pneumonia alba), immune complex mediated nephrotic syndrome, lymphadenopathy, ocular findings (retinitis, uveitis or cataracts), pancreatitis and myocarditis may be present with less frequency.

### 4.2. Late Congenital Syphilis

Late onset disease is seen in children older than 2 years of age and is not contagious. The features of late onset CS are the consequence of protracted inflammation of organs and tissues resulting in permanent scars or destruction of affected organs [21,22]

Dental manifestations are seen in permanent teeth; Hutchinson’s teeth are small, widely separated, barrel shaped and notched; Mulberry molars refer to the presence of many small cusps on the first lower molar instead of the usual four cusps. Treatment of CS in utero or during the first 3 months of life prevents dental changes.

Interstitial keratitis usually appears in puberty, severe inflammation usually presents in one eye later becoming bilateral; it is not seen in children who received treatment before 3 months of age. Eight nerve deafness occurs in only 3% of patients with late CS and typically presents in the first decade of life. Spirochetes are not found in children with deafness or keratitis, and these may respond to steroids. The simultaneous presence of interstitial keratitis, eighth nerve deafness and Hutchinson’s teeth is known as Hutchinson’s triad.

Rhagades are linear scarring lesions radiating from the mouth [23]. Saddle nose is the late sequela of cartilage and bone destruction in children with CS rhinitis. Saddle nose is often associated with failure of the maxilla to grow fully, with relative protuberance of the mandible and associated high palatal arch.

Skeletal sequelae including frontal bossing, saber shins and thickening of the sternal end of the clavicle are the consequence of protracted periostitis. Symmetric, chronic, painless swelling of the knees (Clutton joints) is a rare manifestation of CS.

Incidence of neurological manifestations in CS varies in different reports but may affect as much as a third of patients with abnormal findings in spinal fluid. Developmental delay, hydrocephalus, seizures and cranial nerve involvement have been reported but are rare now.

Late CS is rare in the U.S. these days and represents a failure to diagnose and treat CS early. This is more common in asymptomatic neonates in who the risks or infectious status of the mother were not well evaluated.

## 5. Diagnostic Tests and Management

Definitive diagnosis of CS is made when spirochetes are identified by darkfield microscopy examination of the body fluids or tissue, however, most clinical settings lack the capacity to perform direct detection. *T. pallidum* can be detected by polymerase chain reaction assays, but to date no assays are approved by the U.S. Food and Drug Administration (FDA) for clinical diagnosis. Direct fluorescent antibodies (DFAs) were historically used but are no longer available in the U.S. Instead, most clinical laboratories utilize serological testing to infer a diagnosis of CS since maternal non-treponemal and treponemal IgG antibodies are transferred trans-placentally to the fetus. The finding of an infant’s serum quantitative non-treponemal titer that is four-fold higher than the maternal titer is confirmatory for CS. However, the absence of such finding does not exclude the diagnosis as most infants with CS have titers that are equal to or less than the maternal titer [24,25].

Two algorithms of serological testing are commonly used for the diagnosis of CS, and both require two-step testing (see Figure 4). These algorithms utilize both non-treponemal and treponemal serological tests and only vary in the order in which the tests are performed [16,24,26]. The same non-treponemal tests must be performed on mother and infant so that an accurate comparison can be made. The traditional algorithm starts with a quantitative non-treponemal test such as the rapid plasma reagin (RPR) or Venereal Disease Research Laboratory (VDRL) followed by a confirmatory treponemal-specific test such as fluorescent treponemal antibody absorption (FT-ABS) or *T pallidum* particle agglutination test (TP-PA) if the non-treponemal test is positive. With development of high-throughput automated treponemal-specific immunoassays, many clinical laboratories have opted for the reverse sequence algorithm (RS) for CS screening. The RS starts with a treponemal test such as the treponemal enzyme immunoassay (EIA) or chemiluminescence immunoassay (CIA). If positive, then a quantitative non-treponemal test such as RPR is performed, which, if also positive, it confirms a diagnosis of syphilis in the mother. Then a comparison is made in between the non-treponemal titers of the mother and infant to determine the diagnosis of confirmed or probable CS. If, however, the non-treponemal test is negative then a second treponemal test (TP-PA is the preferred) is performed. If the second treponemal test is negative, then the initial positive result is deemed as a false-positive. However, if the second treponemal test is positive then the diagnosis of previous or current syphilis infection is made on the mother [27,28]. Infants born to women who have a previous history of adequate treatment for syphilis, longer than 4 weeks before delivery and evidence of four-fold declining non-treponemal titer, require no further evaluation if appropriate follow-up of the infant is likely.

All Infants born to mothers with reactive serological tests for syphilis require quantitative non-treponemal testing and should be carefully examined for signs of CS if the mother has no history of adequate treatment. Moreover, given the high rate of co-infection with HIV, these women should be screened for HIV infection as well. If the mother is unavailable for testing, the newborn must be screened utilizing HIV DNA PCR. Appropriate prophylaxis or treatment should be considered. In neonates who have a normal physical examination and a serum quantitative non-treponemal serologic titer that is less than four-fold the maternal titer, evaluation and treatment depends on maternal treatment history. If the mother has untreated syphilis or the treatment is undocumented or inadequate (see above), a complete evaluation consisting of cerebrospinal fluid (CSF) analysis, long bone radiographs, liver enzymes and bilirubin and complete blood cell (CBC) and platelet counts should be performed to guide optimal therapy. If the evaluation is completely normal but follow-up is questionable, the infant should be treated with a single intramuscular dose of benzathine penicillin G [25]. Alternatively, if any abnormality is encountered, the infant should receive a 10-day course of parenteral treatment with crystalline penicillin G or procaine penicillin [3].

Congenital neurosyphilis (CN) is a very difficult diagnosis to derive as most babies with CS have a normal physical and neurological examination. CN can be suspected based on CSF findings, which include a reactive VDRL test, pleocytosis (defined by the CDC as greater than five white blood cells per microliter, although this value in itself can be found in a normal neonate [29]), and elevated protein content (45 mg/dL; 170 mg/dL if the infant is premature). Hence, if the physical exam, laboratory and radiographic results support the diagnosis of CS, treatment must be inclusive of CN regardless of CSF findings [3,24,25].

## 6. Treatment and Follow-Up

Parenteral penicillin G remains as the only antibiotic capable of preventing MTCT and CS. Pregnant women with syphilis should receive the penicillin regimen appropriate for the stage of infection. Pregnant women who have a history of penicillin allergy must be desensitized and treated with penicillin. The decision to treat an infant is dependent on whether the infant has proven or probable CS, has possible CS, or considered less likely or unlikely to have CS (see Figure 5). The treatment of infants with CS based on the CDC and Red Book guidelines is detailed in Figure 5. Note that if more than one dose of penicillin is missed the entire course should be restarted [3,25].

After discharge, an infant is followed every 2–3 months with repeat nontreponemal titers. Infants with CS who are adequately treated should have a four-fold decrease in non-treponemal titers within 6 months and become non-reactive by 12 months. Uninfected infants who had maternal–fetal transmission of IgG non-treponemal antibodies should be seronegative by 6 months. If the non-treponemal tiers increase four-fold at any time or remain stable after 12 months, the infant must be re-evaluated and re-treated with an additional 10-day course of parenteral penicillin G. If the infant at initial evaluation had a positive VDRL consistent with NS, then a repeat lumbar puncture is performed at 6 months of age [3,30].

## 7. Conclusions

Syphilis infection during pregnancy still represents a worldwide public health problem. If left untreated, it can lead to fetal demise and high neonatal morbidity and mortality. CS can be effectively prevented by prenatal serologic screening of mothers and penicillin treatment of infected women, their sexual partners, and when indicated, their newborn infants. The CDC and the U.S. Preventative Services Task Force (USPSTF) both recommend screening all pregnant women during the first trimester and high-risk mothers at 28 weeks’ gestation and delivery [4,6,16]. A study conducted in Florida and Louisiana, following the guidelines for high risk pregnant women, was able to pick up an additional 5% of CS cases by screening at 28 weeks and delivery for an additional treatment of 30 patients over first trimester screening only [32]. However, despite the available data and clinical knowledge, the rates in the U.S. and worldwide continue to rise and predominantly affect disadvantaged communities that have issues with poverty, substance abuse and lack of health care access. To date, the U.S. does not have a national program to mandate screening all pregnant women for CS, in fact, there are still six states where there is no requirement and only a third of states require a third trimester screening [33]. Worldwide, the WHO continues to implement prenatal screening programs that include syphilis testing coupled with appropriate, prompt penicillin treatment for pregnant women. These programs have been able to demonstrate a decrease in cases of CS and, moreover, are cost effective in comparison to dealing with the long-term sequela of untreated CS [30]. Unfortunately, there are many regions in the world where the infrastructure is not available to scale up syphilis screening programs. To meet these challenges, increased awareness of the risks, evaluation and treatment of syphilis in mothers and newborns is critically important as is the call to educate legislators and policy makers on the needs of disadvantaged populations who often cannot advocate for themselves.

## Figures and Tables

**Figure 1 children-07-00203-f001:**
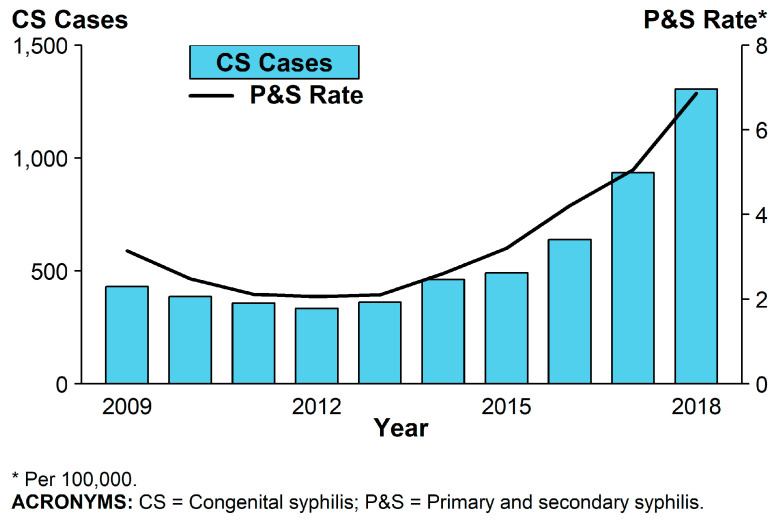
Reported cases by year of birth and rates of reported cases of primary and secondary syphilis among females aged 15–44 years, United States, 2009–2018, adapted from the Centers for Disease Control and Prevention (CDC) *Sexually Transmitted Disease Surveillance 2018* [6].

**Figure 2 children-07-00203-f002:**
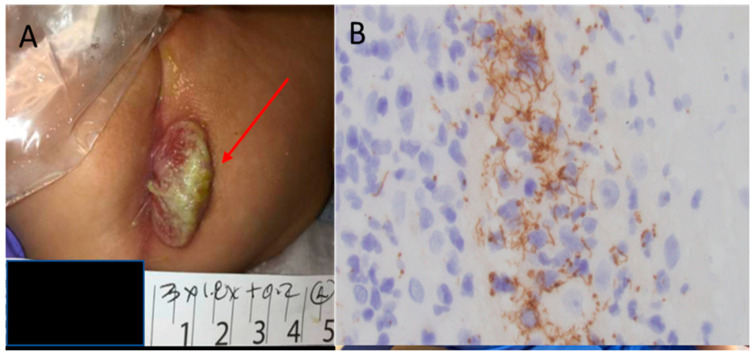
(**A**) Perirectal lesion on a 6 months-old infant with an rapid plasma reagin (RPR) of 1:512. (**B**) Histological staining of a biopsy from the perirectal mass demonstrates a high amount of spirochetes [19]. From Arrieta, A.C. and J. Singh, Congenital Syphilis. 381 (22): p. 2157–2157. Copyright © 2019 Massachusetts Medical Society. Reprinted with permission.

**Figure 3 children-07-00203-f003:**
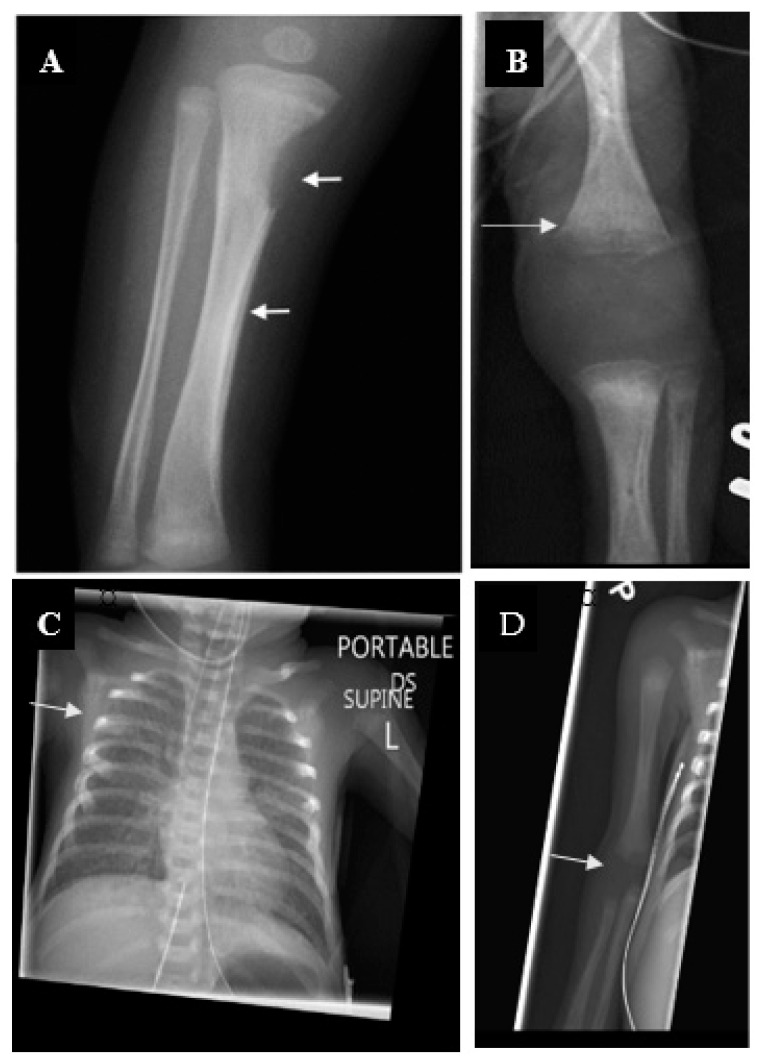
Neonate born to mother without prenatal care highlights importance of testing of pregnant women without prenatal care (PNC) and complete evaluation as not all tests are abnormal. Patient was born at 29 weeks’ gestation to a G7 P3033 homeless woman without PNC. Tested positive for methamphetamine. No hepatosplenomegaly was identified on the exam she had at a + RPR (1:16); positive *T pallidum* particle agglutination test (TP-PA). Her newborn infant had a + RPR (1:256). Her skeletal survey showed extensive periostitis and destruction of the distal metaphysis of both femurs. Her white blood cell count (WBC) was 66.2 K/UL; Hgb 9.1 g/dL. Liver enzymes (AST = 77/ALT = 10) were normal, alkaline phosphatase = 322 U/L, direct bilirubin = 0.3. CSF had a protein of 201 mg/dL, glucose 57 mg/dL, WBC = 1/mm^3^, Venereal Disease Research Laboratory (VDRL) + Ophthalmological exam showed interstitial keratitis. Wimberger sign (**A**). Periostitis of the long bones (**B**). Rib notch associated with congenital syphilis (CS) (**C**). Epiphyseal dislocation with pseudoparalysis of the affected limb (**D**).

**Figure 4 children-07-00203-f004:**
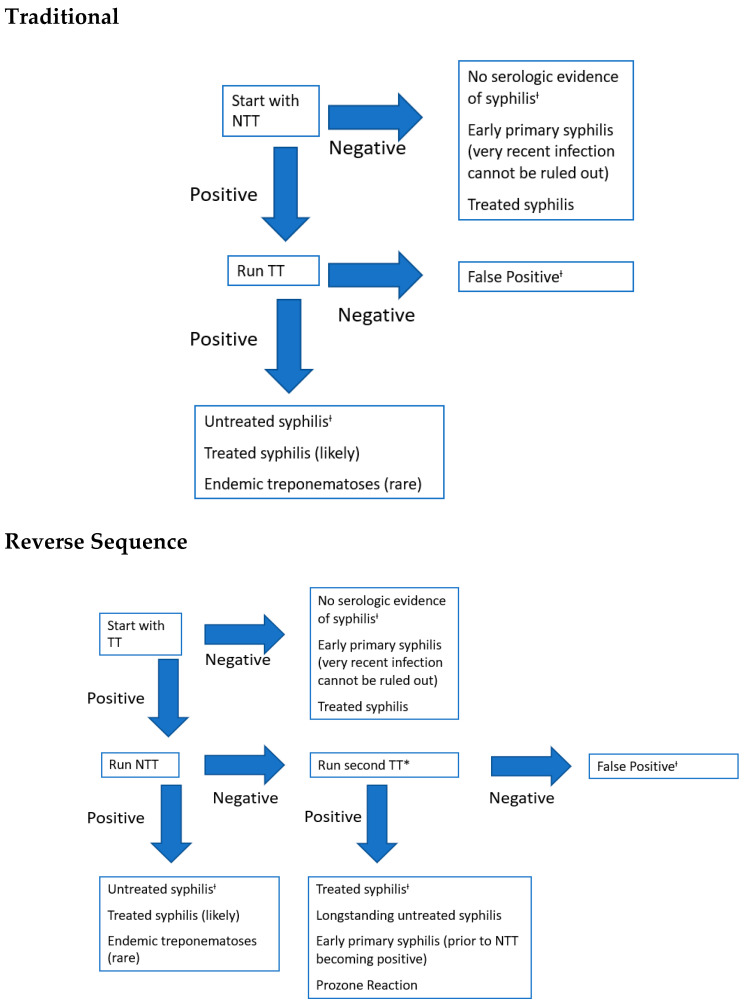
Traditional and reverse sequence algorithms for maternal serologies. NTT = Non-treponemal test. TT = Treponemal test. * = The confirmatory TT must be different from the initial TT. ^Ɨ^ = The likely or most likely interpretation of the test results.

**Figure 5 children-07-00203-f005:**
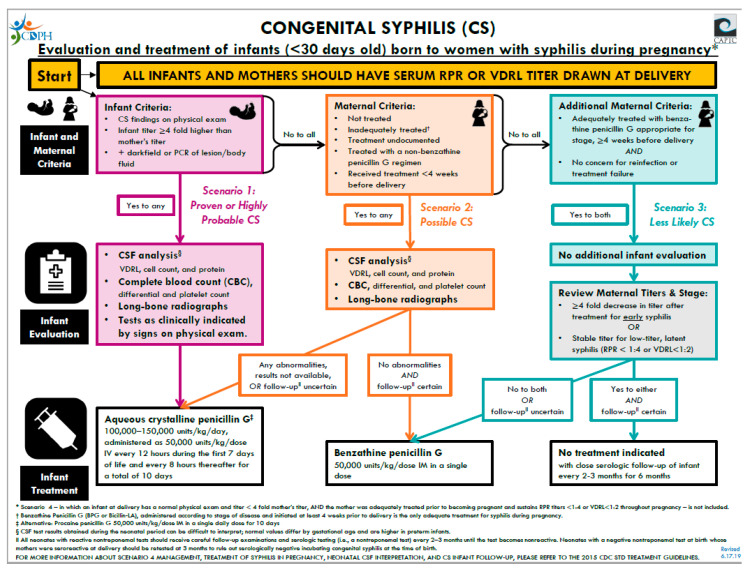
Algorithm for treatment of congenital syphilis. Figure produced by the California Department of Public Health [31].

**Table 1 children-07-00203-t001:** Clinical manifestations of early and late congenital syphilis.

Early Congenital Syphilis	Late Congenital Syphilis
Prenatal		Dentition	
	Nonimmune hydrops		Hutchinson’s teeth
	Intrauterine growth retardation		Mulberry molars
	Stillbirth	Eye	
Hematological/Reticuloendothelial			Interstitial keratitis
	Hepatosplenomegaly	Ear	
	Lymphadenopathy		Eight nerve deafness
	Thrombocytopenia	Nose/Face	
	Anemia		Saddle nose
	Leukopenia/Leukocytosis (monocytosis)		Impaired maxillary growth
Mucocutaneous		Cutaneous	
	Rhinitis (sniffles)		Rhagades
	Rash (papulovesicular, prominent in palms and sole; pemphigus syphiliticus)	Skeletal	
	Mucous patches		Frontal bossing
Skeletal			Saber shins
	Long bone lesions (Wimberger sign in tibial protuberance)		Clavicle (sternal end) hypertrophy (Higoumenakis’ sign)
	Periostitis (pseudoparalysis)		Clutton’s joints (knees)
Neurological		Neurological	
	Aseptic meningitis		Aseptic meningitis/asymptomatic neurosyphilis
Ocular			
	Retinitis (rare, also cataracts and keratitis)

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
