# Peer review of "Congenital Syphilis: A U.S. Perspective"

_children, 2020, doi:10.3390/children7110203_

Round 1

Reviewer 1 Report

Reviewer's suggestions

Short summary

This article reviews the concept, epidemiology (in the United States), biology, clinical manifestations, management for diagnosis, treatment, and follow-up of Congenital Syphilis. The manuscript is clear and concise, using very adequate bibliographic references. The figures and the table are illustrative, educational and useful for clinical practice.

Specific Comments

Line 22: … gestation; risk of … (semicolon)

Line 55: … interventions: a) reach … (colon)

Line 92: 4.3. Clinical Manifestations (4)

Line 102: 4.3.1. Early Congenital Syphilis. (4)

Table 1: Wimberger (no Wienberger)

Line 126: Wimberger

Line 129: … is common; cerebrospinal … (semicolon)

Line 145: Wimberger

Line 146: Remove the “point” (.) before the (C)

Line 147: 4.3.2. Late Congenital Syphilis. (4)

Line 174: 5. Diagnostic Tests … (5)

Line 202: [27, 28] .. Infants (remove one point)

Line 234: 6. Treatment and … (6)

Line 250: 7. Conclusion (7)

Put the “point” (.) at the end of the sentence:

  • Lines: 13, 30, 35, 41, 53, 60, 66, 78, 82, 91, 99, 101, 105, 128, 150, 184, 187, 226, 239, 249, 258.

Remove the “point” (.) before the [ ]:

  • Lines: 78, 91, 105, 117, 184, 187, 226.

References: adapt all references to ACS style guide.

Reviewer 2 Report

This is a nice review article on CS that highlights some of the important issues related to this manifestation of syphilis.

some minor changes are suggested below:

  1. line 17. When the pathogen is mentioned for the first time the subspecies should be reported (e.g. T. pallidum subsp. pallidum).
  2. line 76. endemicum, not endemiccum
  3. line 71. Pallidum should not be capital letter
  4. line 81. the endoflagellum confers motility and shape, what other organelles are the authors referring to that are important for motility?
  5. in general, rather than giving a generic description of the agent in the "Biology of Treponema pallidum" paragraph, the authors should try and report briefly on virulence factors that might be involved in placental adhesion and invasion. The literature is scarce, but it would be more pertinent to this work. this is among the few bacteria that can cross the placenta, and the biological basis for it are still largely lacking, although some works point at putative surface-exposed lipoproteins as possible placental adhesins.
  6. table 1 should not be split between two pages but be in one page only
  7. punctuation should be checked throughout
